# DiffVL: Scaling Up Soft Body Manipulation using Vision-Language Driven Differentiable Physics

**Zhiao Huang**[*]
Computer Science & Engineering
University of California, San Diego
z2huang@ucsd.edu

**Feng Chen**[*]
Institute for Interdisciplinary Information Sciences
Tsinghua University
chenf20@mails.tsinghua.edu.cn

**Yewen Pu**
Autodesk
yewen.pu@autodesk.com

**Chunru Lin**
UMass Amherst
chunrulin@umass.edu

**Hao Su**[†]
University of California, San Diego
haosu@ucsd.edu

**Chuang Gan**[†]
MIT-IBM Watson AI Lab, UMass Amherst
chuangg@umass.edu

## Abstract

Combining gradient-based trajectory optimization with differentiable physics simulation is an accurate and efficient technique for solving soft-body manipulation problems. Using a well-crafted optimization objective, the solver can quickly converge onto a valid trajectory. However, writing the appropriate objective functions requires expert knowledge, making it difficult to collect a large set of naturalistic problems from non-expert users. We introduce DiffVL, a framework that integrates the process from task collection to trajectory generation leveraging a combination of visual and linguistic task descriptions. A DiffVL task represents a long horizon soft-body manipulation problem as a sequence of 3D scenes (key frames) and natural language instructions connecting adjacent key frames. We built GUI tools and tasked non-expert users to transcribe 100 soft-body manipulation tasks inspired by real-life scenarios from online videos. We also developed a novel method that leverages large language models to translate task language descriptions into machine-interpretable optimization objectives, which can then help differentiable physics solvers to solve these long-horizon multistage tasks that are challenging for previous baselines. Experiments show that existing baselines cannot complete complex tasks, while our method can solve them well. Videos can be found on the website https://sites.google.com/view/diffvl/home.

## 1 Introduction

This paper focuses on soft body manipulation, a research topic with a wide set of applications such as folding cloth [52, 90, 34], untangling cables [88, 87], and cooking foods [77, 72]. Due to their complicated physics and high degree of freedom, soft body manipulations raise unique opportunities and challenges. Recent works such as [32] and [99] have heavily leveraged various differentiable physics simulators to make these tasks tractable. Towards generalizable manipulation skill learning, such approaches have the potential to generate data for learning from demonstration

---

[*]Equal contribution
[†]Equal contribution

37th Conference on Neural Information Processing Systems (NeurIPS 2023).

algorithms [51, 43]. However, to enable differentiable physics solvers to generate a large scale of meaningful trajectories, we must provide them with suitable tasks containing the scene and the optimization objectives to guide the solver. Previous tasks are mostly hand-designed [32] or procedure-generated [51], resulting in a lack of diverse and realistic soft-body manipulation tasks for researchers.

This paper takes another perspective by viewing tasks as data, or more precisely, task specifications as data. Each data point contains a pair of an initial scene and an optimization objective depicting the goal of the task. Taking this perspective, we can annotate meaningful tasks like annotating data of other modalities, like texts, images, videos, or each action trajectory, by leveraging non-expert human labor, providing possibilities for scaling up the task space of differentiable physics solvers.

A core problem in building a task collection framework with trajectory generation is finding a suitable representation of the tasks. The representation should be intuitive for non-expert annotators while being accurate enough to describe the complex shapes and motions of the soft bodies that appear in various soft body manipulation processes. Last but not least, it should be reliably interpreted by differentiable physics solvers to yield a valid trajectory.

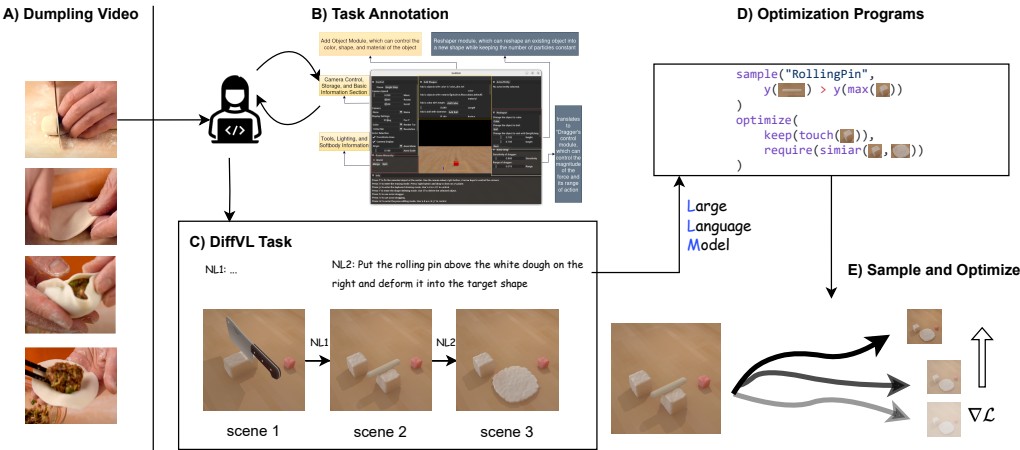

Figure 1: (A) A dumpling making video; (B) The annotator interacts with our GUI tool to create DiffVL tasks; (C) A DiffVL task contains a sequence of 3D scenes along with natural language instructions to guide the solver; (D) DiffVL leverages a large language model to compile instructions into optimization programs consisting of vision elements; (E) The optimization program guides the solver to solve the task in the end.

Studies such as [3, 1, 60, 89] have shown that humans are proficient at defining goal-driven, spatial object manipulation tasks using either sketches, natural languages, or both. Figure 1(A), for instance, depicts the complete process of dumpling creation, supplemented by textual instructions. Taking this as an inspiration, we present DiffVL, a framework that enables non-expert users to specify soft-body manipulation tasks to a differentiable solver using a combination of vision and language. Specifically, each DiffVL task consists of a sequence of 3D scenes (keyframes), with natural language instruction connecting adjacent keyframes. The sequence of key frames specifies the sequence of *subgoals* of the manipulation task, and the natural language *instructions* provides suggestions on how to use the actuators to manipulate the objects through this sequence of subgoals. See Figure 1.

We develop tools to ease the annotation for non-expert users. With our interactive simulator equipped with a GUI, the user can edit, manipulate, draw, or carve shapes easily like other 3D editing tools and observe the consequence through simulation in an intuitive manner. Meanwhile, when it is tedious for users to edit all the intermediate steps of a complex motion, they can use natural language to describe the goal instead of drawing them step by step. This enables us to build **SoftVL100**, a vision-language dataset with 100 diverse tasks. DiffVL uses LLM to compile the natural language instructions to an optimization program – which enforces a set of constraints during the soft-body manipulations from one keyframe to the next. This optimization program is then used by a differentiable physics solver to generate a working trajectory.

To summarize, our work makes the following contributions:

- We propose a new multi-stage vision-language representation for defining soft-body manipulation tasks that is suitable for non-expert user annotations.
- We developed a corresponding GUI, and curated SoftVL100, consisting of 100 realistic soft-body manipulation tasks from online videos[3].
- We develop a method, DiffVL, which marries the power of a large-language model and differentiable physics to solve a large variety of challenging long-horizon tasks in SoftVL100.

## 2   Related Work

**Differentiable simulation for soft bodies** Differentiable physics [30, 93, 69, 19, 29] has brought unique opportunities in manipulating deformable objects of various types [58, 28, 53, 33, 92, 49, 95, 83, 63, 100, 95, 56, 27, 47, 74, 23], including plasticine [32], cloth [45], ropes [54] and fluids [96] simulated with mass-spring models [30], Position-based Dynamics [65], Projective Dynamics [69, 16], MPM [36] or Finite Element Method [26], which could be differentiable through various techniques [30, 16, 57, 7]. It has been shown that soft bodies have smoother gradients compared to rigid bodies [32, 30, 93, 6, 82], enabling better optimization. However, the solver may still suffer from non-convexity [44] and discontinuities [82], motivating approaches to combine either stochastic sampling [44], demonstrations [43, 6] or reinforcement learning [98, 64] to overcome the aforementioned issues. DiffVL capitalizes on an off-the-shelf differentiable physics solver to annotate tasks. It incorporates human priors through a novel vision-language task representation. This allows users to employ natural language and GUI to guide the differentiable solver to solve long-horizon tasks, distinguishing DiffVL from previous methods.

**Task representation for soft body manipulation** There are various ways of defining tasks for soft body manipulation. To capture the complex shape variations of soft bodies, it is natural to use either RGB images [95, 51, 73] or 3D point cloud [78, 47] to represent goals. However, given only the final goal images, it may be hard for the solver to find a good solution without further guidance. Besides, generating diverse images or point clouds that adhere to physical constraints like gravity and collision avoidance in varied scenes can be challenging without appropriate annotation tools. Our GUI tools are designed to address these issues, simplifying the process. Other works use hand-defined features [10, 88] or formalized languages [41]. These methods necessitate specialist knowledge and require tasks to be defined on a case-by-case basis. Learning from demonstrations [43, 73] avoids defining the task explicitly but requires agents to follow demonstration trajectories and learn to condition on various goal representations through supervised learning or offline reinforcement learning [20, 18, 79, 24, 55, 35, 11, 2, 5]. However, collecting demonstrations step-by-step [70, 101, 43] requires non-trivial human efforts, is limited to robots with human-like morphologies, and may be challenging for collecting tasks that are non-trivial for humans (e.g., involving complex dynamics or tool manipulation). Our annotation tool emphasizes the description of tasks over the identification of solutions. Annotators are merely required to define the task, not execute the trajectories themselves, thereby simplifying the annotation process. Our method is also related to [46], but we consider a broader type of tasks and manipulation skills.

**Language-driven robot learning** Many works treat language as subgoals and learn language-conditioned policies from data [80, 21, 66, 42, 8, 62, 61] or reinforcement learning [37, 17, 84, 12]. It has been shown that conditioning policies on languages lead to improved generalization and transferability [80, 37]. [76] turns languages into constraints to correct trajectories. Our method distinguishes itself by integrating language with differentiable physics through machine-interpretable optimization programs. Recent works aim at leveraging large language models [4, 15, 94, 50, 13] to facilitate robot learning. Most use language or code [48] to represent policies rather than goals and focus on high-level planning while relying on pre-defined primitive actions or pre-trained policies for low-level control. In contrast, our language focuses on low-level physics by compiling language instructions to optimization objectives rather than immediate commands. It's noteworthy that VIMA [38] also introduces multimodal prompts, embodying a similar ethos to our vision-language task representation. Their approach views multimodal representation as the task prompts to guide the policy. However, it still requires pre-programmed oracles to generate offline datasets for learning, which is hard to generalize to soft-body tasks with complex dynamics. Contrastingly, we

---

[3]both the GUI and dataset will be made public

adopt a data annotation perspective, providing a comprehensive framework for task collection, natural language compiling, and trajectory optimization, enabling us to harness the potential of differentiable physics effectively.

**Task and motion planning** Our optimization program compiled from natural language is closely connected to the field of Task and Motion Planning (TAMP) [22], particularly Logic Geometric Programming (LGP) [85, 86]. We draw inspiration from LGP's multiple logical states in designing our DiffVL task representation. However, unlike LGP, our approach asks human annotators to assign "logic states" and facilitates the manipulation of soft bodies through our novel vision-language representation. Some recent works have explored the intersection of TAMP and language models [14, 81, 59] while primarily focusing on generating high-level plans.

## 3  SoftVL100: Vision-Language Driven Soft-body Manipulation Dataset

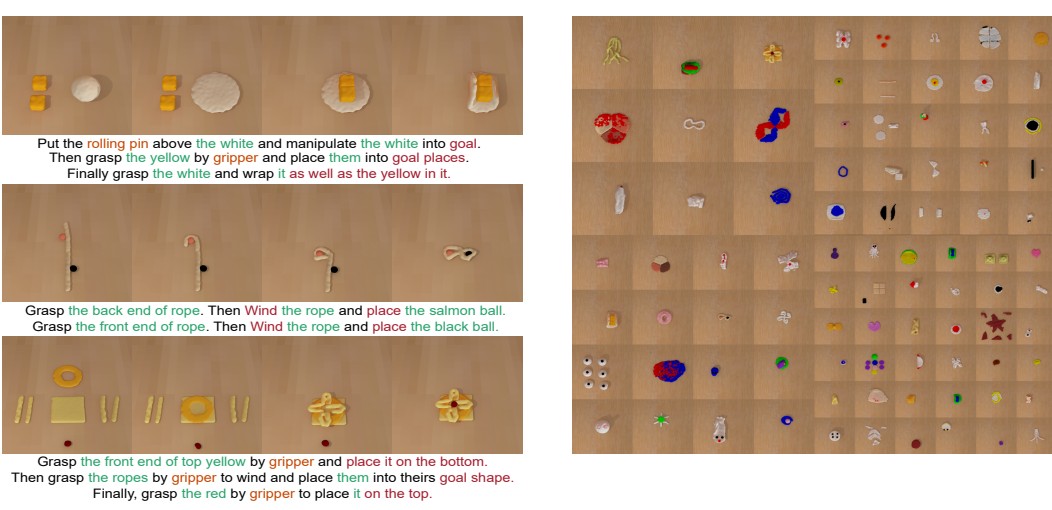

(A)Multistage Tasks

(B) Scene examples

Figure 2: (A) Example of multi-stage tasks and their text annotations; (B) Snapshots of scenes in SoftVL100 dataset.

In this representation, the original tasks are divided into a sequence of 3D scenes, or key frames, along with text descriptions that detail the steps for progressing to the next scene. Figure 2(A) illustrates our representation. Each key frame consists of multiple soft bodies with different positions and shapes. By grouping two consecutive key frames together, we form a manipulation stage. Natural language instructions may guide the selection of an actuator and provide guidance on how to control it to reach the subsequent key frames. In Section 3.1, we develop annotation tools to facilitate the task representation annotation process. These tools make it easy to construct a diverse set of tasks, referred to as SoftVL100, as discussed in Section 3.2.

### 3.1  Vision-Language Task Annotator

Our annotation tool is based on PlasticineLab [32], a simulation platform that utilizes the Material Point Method (MPM) [31] for elastoplastic material simulation. To enhance the user experience and support scene creation, we integrate it into SAPIEN [97], a framework that enables the development of customized graphical user interfaces.

Our simulation runs on GPU servers and the interface is accessible as a web service through VNC, allowing users to interact with the simulator directly from their web browser. When

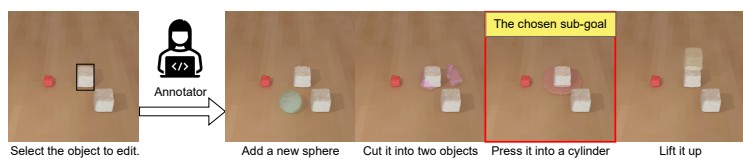

Figure 3: Example operations in GUI tools

starting a new task annotation, users can open the simulator, perform actions, and save the current scene. Our GUI tools provide comprehensive support for creating, editing, managing, and simulating objects, as illustrated in Figure 3. **Scene creation:** The GUI offers a variety of primitive shapes, such as ropes, spheres, cubes, and cylinders, that users can add to the scene. Before adding a soft-body shape, users can adjust its size, color, and materials (e.g., rubber, fiber, dough, iron) if needed. **Shape editing:** Users have the ability to select, edit, or delete shapes within the scene. The GUI tool includes a range of shape operators, including moving, rotating, carving, and drawing. We provide a complete list of supported operations in the appendix. **Simulation:** Since the interface is built on a soft body simulator, users can simulate and interact with the soft body using rigid actuators or magic forces. They can also run simulations to check for stability and collision-free states. **Object management:** Users can merge connected shapes to create a single object or divide an object into two separate ones with distinct names. The annotation tool keeps tracking the identities of objects across scenes.

Once users have finished editing the scene and creating a new key frame, they can save the scene into the key frame sequence. The key frames are displayed below the simulator, visualizing the current task annotation progress. Users have the option to open any of these saved scenes in the simulator, delete a specific scene if needed, and add text annotations for each scene. Please refer to the appendix for detailed information on the functionalities and usage of the GUI tool.

### 3.2 The SoftVL100 Dataset

Our annotation tool simplifies the task creation process, enabling us to hire non-expert users to collect new tasks to form a new task set, SoftVL100. The task collection involves several stages, starting with crawling relevant videos from websites like YouTube that showcase soft body manipulation, particularly clay-making and dough manipulation. We developed tools to segment these videos and extract key frames to aid in task creation.

We hired students to annotate tasks, providing them with a comprehensive tutorial and step-by-step guidelines for using our annotation tools. It took approximately two hours for each annotator to become proficient in using the annotation tool. Subsequently, they were assigned a set of real-world videos to create similar tasks within the simulator. After collecting the keyframes, we proceed to relabel the scenes with text descriptions. Annotators are provided with a set of example text descriptions. We encourage users to include detailed descriptions of the actuators, their locations, moving directions, and any specific requirements, such as avoiding shape breakage for fragile materials. The annotation process for each task typically takes around 30 minutes. Using our task annotation tool, we have created a dataset called SoftVL100, which consists of 100 tasks, and there are more than 4 stages on average. The tasks cover a wide range of skills as illustrated in Figure 2 (A). Sample tasks from the dataset can be seen in Figure 2 (B).

## 4 Optimization with Vision-Language Task Description

We propose DiffVL to tackle the challenging tasks in SoftVL100. Given a stage of a task depicted in Figure 4(A), DiffVL utilizes large language models to compile the natural language instructions into a machine-interpretable optimization program in Figure 4(B). The program comprises of the names of visible elements and Python functions, effectively capturing the essence of the language instructions. We introduce the design of our DSL in Section 4.1 and outline the DiffVL compiler based on large language models (LLM) in Section 4.2. The resulting optimization program includes crucial information for selecting and locating actuators and can facilitate a differentiable physics solver to generate valid trajectories, as discussed in Section 4.3.

### 4.1 Optimization Program

The optimization program is formulated using a specialized domain-specific language (DSL) that incorporates several notable features. Firstly, it includes functions that extract the names of visible elements from the 3D scenes and support operations on these elements, represented as 3D point clouds. This enables manipulation and analysis of the visual information within the optimization program. Secondly, the DSL provides various functions to express geometric and temporal relations, encompassing common geometric and motion constraints. These functions facilitate the representation and handling of natural language instructions. Furthermore, the DSL clauses are interpreted

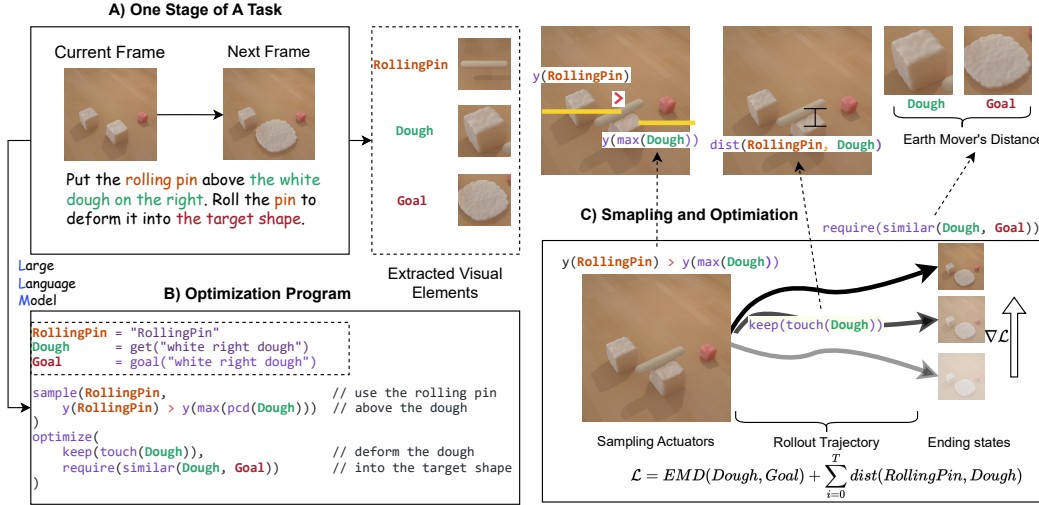

Figure 4: (A) One stage of the DiffVL task of two keyframes and a natural language instruction. (B) The compiled optimization program. (C) The sampling and optimization process.

into PyTorch [68], making the constraints automatically differentiable and directly applicable for differentiable physics solvers.

The example in Figure 4 already exposes multiple components of our DSL. In the program, we can refer to the names of visible elements through `get` and `goal` functions, which can look for and extract the 3D objects that satisfy the description, identified by their color, shapes, and locations (we ensured that these attributes are sufficient to identify objects in our tasks). The program will then start with a `sample` statement, which is used to specify the actuators and their associated constraints over initial positions. Actuators refer to the entities utilized for manipulation, such as a robot gripper, a knife, or the rolling pin depicted in Figure 4. The constraint `y(RollingPin) > y(max(pcd(Dough)))` within the `sample` statement represents a specific constraint. It ensures that the height of the actuator (RollingPin) is greater than the highest $y$ coordinate of the white dough in the scene, reflecting the meaning of "put bove" of the language instructions. Following the `sample` statement, an `optimize` statement is included. This statement plays a vital role in defining the constraints and objectives for the manipulation trajectories. These constraints and objectives serve as the differentiable optimization objectives that the differentiable physics solver aims to maximize. In an optimization program, two special functions, namely `require` and `keep`, can be used to specify the temporal relationship of the optimization objectives. By default, the `require` function evaluates the specified condition at the end of the trajectories. On the other hand, the `keep` function is applied to each frame of the trajectories, ensuring that the specified conditions are maintained throughout the trajectory. In the example program shown in Figure 4(B), the optimization program instructs the actuator to continuously make contact with the white dough. The function `similar` computes the Earth Mover distance [71] between two objects' point clouds, which aims to shape the dough into a thin pie, the goal shape extracted from the next key frame. We illustrate more example optimization programs in Figure 5.

## 4.2 Compiling Natural Languages with LLM

We utilize the power of the Large Language Model (LLM) to compile text instructions into optimization programs. This approach capitalizes on the few-shot capabilities of the LLM and employs various few-shot prompts [91, 9]. The prompts begin by defining the "types" and "functions" within our DSL. Each function is accompanied by a language explanation to clarify its purpose. Additionally, examples are given to demonstrate how primitive functions can be combined to form complex clauses. To facilitate the translation, we provide the language model with both the scene description of the

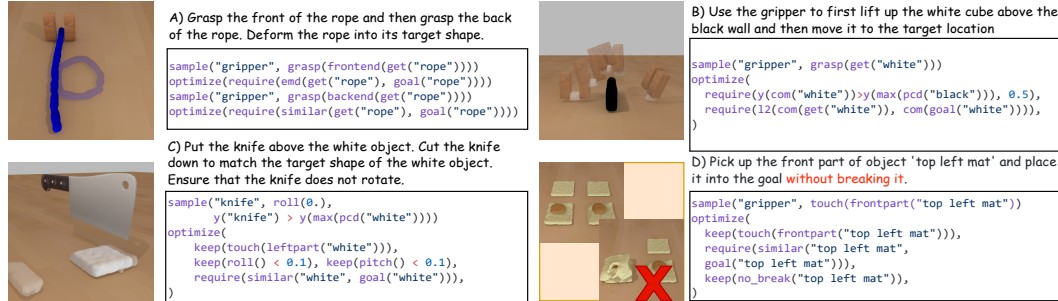

Figure 5: A) Decomposing a stage into sub-stages: manipulating a rope into a circle using a single gripper requires the agent to first grasp the upper part of the rope and then manipulate the other side to form the circle. The program counterpart contains two pairs of `sample` and `optimize`, reflecting the two sub-stages in the language instruction. `backend` extracts the back end of 3D objects; B) Guiding the motion of the objects to avoid local optima: we use `com` to compute the center of the mass of objects and `pcd` to get the clouds; relation operators compare two objects' coordinate. The first `require` takes an additional argument $0.5$, meaning the inner clause, which asks the white object above the black one, should be evaluated halfway through the trajectory to represent meanings of 'first do A and then do B.' C) Selecting a suitable tool to split the objects; D) The program can include additional constraints like not breaking the shape, which is critical for manipulating fragile materials.

objects that it contains and the natural language instructions that we wish to compile. We then prompt the model with the instruction, "Please translate the instructions into a program." This step is repeated for all frames, resulting in their respective optimization programs. Through our experimentation, we have observed that GPT-4 [67] surpasses other models in terms of performance. Please refer to the appendix for a more comprehensive understanding of the translation process.

### 4.3 Solving the Optimization through Differentiable Physics

We leverage the differentiable physics solver in [32], which has been proven accurate and efficient in solving the generated optimization programs. The process is illustrated in Figure 4(C). The `sample` function is employed within a sampling-based motion planner. Initially, it samples the pose of the specified actuator type to fulfill the provided constraints. Subsequently, an RRT planner [40] determines a path toward the target location, and a PD controller ensures that the actuators adhere to the planned trajectory. With the actuator suitably initialized, the `optimize` clause is then passed to a gradient-based optimizer to optimize the action sequence for solving the task. The solver performs rollouts, as depicted in Figure 4(C), where the conditions enclosed by `keep` operators are evaluated at each time step. Conversely, conditions enclosed by `require` are evaluated only at specified time frames. The resulting differentiable values are accumulated to calculate a loss, which is utilized to compute gradients for optimizing the original trajectories. For handling multistage tasks involving vision-language representation, we can solve them incrementally, stage by stage.

## 5 Experiments

In this section, we aim to justify the effectiveness of our vision-language task representation in guiding the differentiable physics solver. For a detailed analysis of the ablation study on the language translator, we direct readers to the appendix. To better understand the performance of different solvers, we evaluate the tasks of two tracks. The first track focuses on tasks that only require agents to transition from one key frame to another. Tasks in this track are often short-horizon. This setup allows us to evaluate the efficiency and effectiveness of the differentiable physics solver in solving short-horizon soft body manipulation tasks, as described in Section 5.2. In the second track, we validate the agent's capability to compose multiple key frames in order to solve long-horizon manipulation tasks.

These tasks involve complex scenarios where agents need to switch between different actuators and manipulate different objects. The evaluation of this composition is discussed in Section 5.3.

## 5.1 Initialization for tasks

We clarify that our approach does not use a gradient-based optimizer to optimize trajectories generated by the RRT planner. Motion planning and optimization occur in separate temporal phases. Initially, we plan and execute a trajectory to position the actuator at the specified initial pose. Subsequently, we initialize a new trajectory starting from this pose, and the actuator remains fixed in this trajectory as we initialize the policy with zero actions to maintain its position. The sample function generates poses in line with the annotators' guidance.

## 5.2 Mastering Short Horizon Skills using Differentiable Physics

Before evaluating the full task set, we first study how our vision-language representation can create short-horizon tasks that are solvable by differentiable physics solvers. We picked 20 representative task stages as our test bed from the SoftVL100. We classify these tasks into 5 categories, with 4 tasks in each category. The categories are as follows: **Deformation** involves the deformation of a large single 3D shape, achieved through pinching and carving; **Move** asks the movement of an object or stacking one object onto another; **Winding** involves the manipulation of a rope-like object to achieve different configurations. **Fold** focuses on folding a thin mat to wrap other objects. **Cut** involves splitting an object into multiple pieces. The selected tasks encompass a wide spectrum of soft body manipulation skills, ranging from merging and splitting to shaping. We evaluate different methods' ability to manipulate the soft bodies toward their goal configuration. The agents are given the goal configuration in the next frame, and we measure the 3D IoU of the ending state as in [32]. We set a threshold for each scene's target IoU score to measure successful task completion, accommodating variations in IoU scores across different scenes.

We compare our method against previous baselines, including two representative reinforcement learning algorithms, Soft Actor-Critic (SAC) [25] and Proximal Policy Gradient (PPO) [75]. The RL agents perceive a colored 3D point cloud as the scene representation and undergo $10^6$ steps to minimize the shape distance towards the goal configuration while contacting objects. Additionally, we compare our approach with CPDeform [44], which employs a differentiable physics solver. However, CPDeform uses the gradient of the Earth mover's distance as a heuristic for selecting the initial actuator position. It differs from DiffVL which utilizes the text description to determine ways to initialize the actuators and the optimization objectives. For differentiable physics solvers, we run Adam [39] optimization for 500 gradient steps using a learning rate of 0.02.

Table 1 presents the success rates and mean IoU of various approaches. It is evident that, except for specific deformation and cutting tasks where SAC shows success, the RL baselines struggle to solve most tasks. Although RL agents demonstrate an understanding of how to manipulate the soft bodies correctly, they lack the precision required to match the target shape accurately. Additional experimental results can be found in the appendix. CPDeform achieves partial success in each task category. It surpasses the RL baselines by utilizing a differentiable physics solver and employing a heuristic that helps identify contact points to minimize shape differences. However, CPDeform's heuristic lacks awareness of physics principles and knowledge about yielding objects to navigate around obstacles. Consequently, CPDeform fails to perform effectively in other tasks due to these limitations. We want to emphasize that integrating language into RL baselines entails non-trivial challenges. For example, many reward functions in our optimization program have temporal aspects, making the original state non-Markovian. As we focus on enhancing the optimization baseline through language-based experiments, we've chosen to defer tackling the complexities of language-conditioned RL to future research.

Our approach surpasses all other methods, achieving the highest overall success rate. To examine the impact of different components and illustrate their effects, we conduct ablation studies. First, we remove the constraints in `sample` named "- Sample" in the table and ask the agent to sample random tools without using hints from the language instructions. As a result, the performance drops significantly, either getting stuck at the local optima or failing to select suitable tools to finish the task. For example, in the cutting task, it failed to select knives and split the white material into two pieces. On the other hand, the variant "- Optimize" replaces the statement in `optimize` by

| | previous baselines | | | DiffVL | | |
|---|---|---|---|---|---|---|
| | **SAC** | **PPO** | **CPDeform** | **- Sample** | **- Optimize** | **Ours** |
| | **SR/IOU** | **SR/IOU** | **SR/IOU** | **SR/IOU** | **SR/IOU** | **SR/IOU** |
| Deform | 0.25 (0.504) | 0.00/0.392 | 0.25/0.532 | 0.11/0.489 | 0.44/0.524 | **1.00/0.564** |
| Move | 0.00/0.541 | 0.00 /0.527 | 0.08 /0.570 | 0.00/0.533 | 0.33/0.618 | **1.00/0.641** |
| Wind | 0.00/0.308 | 0.00/0.289 | 0.21/0.362 | 0.25 /0.351 | 0.08 /0.408 | **0.59/0.446** |
| Fold | 0.00/0.572 | 0.00 /0.457 | 0.33/0.612 | 0.00/0.550 | **1.00 /0.650** | 0.94/0.643 |
| Cut | 0.33/0.449 | 0.00/0.408 | **0.89/0.482** | 0.33/0.357 | 0.56/0.447 | 0.87/0.490 |
| Total | 0.12/0.475 | 0.00/0.415 | 0.35/0.512 | 0.14/0.456 | 0.48/0.529 | **0.88/0.557** |

Table 1: Single stage experiment results. The metrics we use are Success Rate (SR) and 3D Intersection Over Union (IOU).

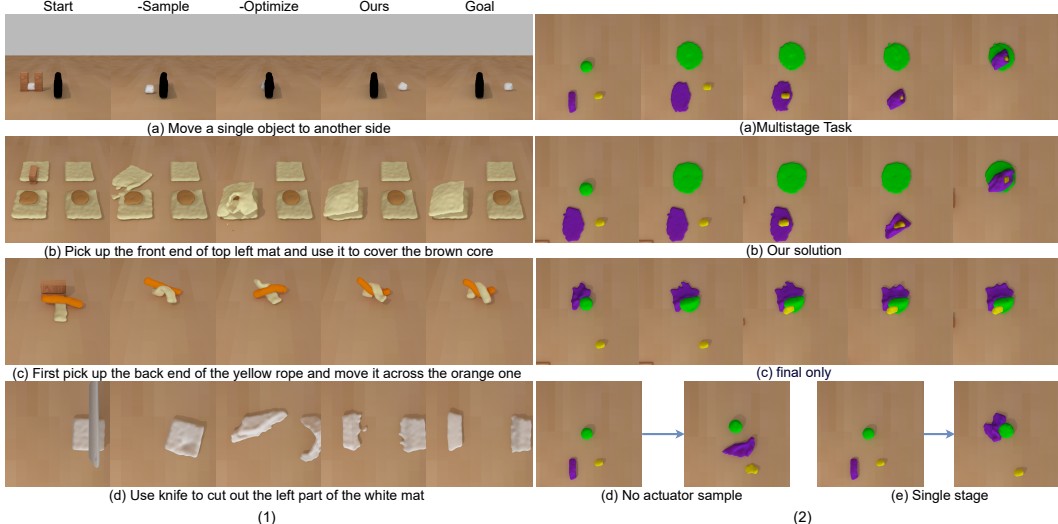

Figure 6: (1) Performance on single stage tasks. (2)Performance on a multistage task that includes moving, packing, and pressing.

an objective focusing solely on optimizing shape distance, removing additional guidance for the physical process. The effects are lesser compared to the previous variant, demonstrating that with a suitable tool initialization method, a differentiable physics solver is already able to solve many tasks. However, as illustrated in Figure 6(A), it does not lift the white cube to avoid the black wall in a moving task and breaks the mat shown in the second row. In the winding task, it fails to find the relatively complex lifting-then-winding plan directly from shape minimization. In contrast, our method can successfully solve these tasks after leveraging the trajectory guidance compiled from the natural language instructions. We find that our solver is quite efficient. For a single-stage task, it takes 10 minutes for 300 gradient descent steps on a machine with NVIDIA GeForce RTX 2080, for optimizing a trajectory with 80 steps. For most tasks, 300 gradient steps are sufficient for finding good solutions.

## 5.3  Driving Multi-Stage Solver using Vision-Language

Having evaluated short-horizon tasks, we apply our method to long-horizon tasks in SoftVL100. We leverage the natural language instructions to generate the evaluation metrics for each task, which might measure the relationships between objects, check if soft materials rupture, or if two shapes are similar. We refer readers to the appendix for more details.

Our multistage vision-language task representation decomposes the long-horizon manipulation process into multiple short-horizon stages, allowing our method to solve them in a stagewise manner. As baseline algorithms fail to make notable progress, we apply ablation to our method: We first compare

our method with a single-stage approach (single), which refers to solving long-horizon tasks like short-horizon tasks, by dropping the key frames and directly optimizing for the final goal. We then dropped the actuator sampling process in the middle (no actuator sample). The solver is still told to optimize for key frames (sub-goals) at each stage but was not informed to switch actuators and objects to manipulate. As expected, dropping the stage information and the language instructions in our vision task representation significantly degrades the performance, making solvers unable to complete most tasks.

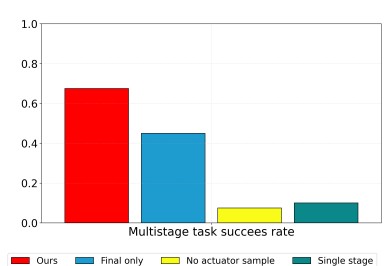

Figure 7: Success rate of multi-stage tasks of different ablations

After leveraging the language instructions, the FinalOnly solver removes the vision subgoals in each sub-stage but provides the agent's actuators to choose the objects to manipulate. In this case, similar to the "- Sample" variant, the differentiable physics solver can solve certain tasks by only minimizing shape distance. However, it may have troubles for tasks that require objects to deform into multiple shapes. For example, as illustrated in Figure 6(B), the solver needs to first compress the purple shape into a thin mat to warp the yellow dough.

The solver fails to discover the compression process after removing the subgoals but directly moves the purple toward its final configurations, resulting in the failure in the end. This showcases the importance of the introduction of the key frames in our vision-language task annotation.

## 6    Limitations and Conclusion

We introduce DiffVL, a method that enables non-expert users to design specifications of soft-body manipulation tasks through a combination of vision and natural language. By leveraging large language models, task descriptions are translated into machine-interpretable optimization objectives, allowing differentiable physics solvers to efficiently solve complex and long-horizon multi-stage tasks. The availability of user-friendly GUI tools enables a collection of 100 tasks inspired by real-life manipulations. However, it is important to acknowledge some limitations. Firstly, the approach relies on human labor for dataset creation. Additionally, the utilization of large language models incurs high computational costs. Nevertheless, We anticipate being able to gather a larger set of tasks in the future. Furthermore, leveraging the generated trajectories to guide learning methods holds significant potential as a promising avenue for future exploration.

## 7    Future Works

We believe that crowdsourcing is vital in achieving this goal. We have taken the first step to demonstrate how non-experts can contribute to scaling up robot learning, which may inspire the field to explore ways of involving individuals from various backgrounds in expanding robot manipulation capabilities. It would be out of the capabilities of our group but can be readily adopted by other groups with better resources eventually.

Transferring the policies found by the differentiable simulator to the real world is important. We believe that various components of our framework can offer valuable contributions to real-world scenarios: one such contribution lies in our vision language representation, which naturally represents tasks in the real world; leveraging large language models aids in generating rewards for evaluating real-world results; our GUI tool can effectively manipulate real-world materials, particularly in constrained tasks, serving as a direct interface between humans and robot controllers.

## 8    Acknowledgements

This work is supported by DSO grant DSOCO2107, MIT-IBM Watson AI Lab, and gift funding from MERL, Qualcomm, and Cisco. The exchange program of Feng Chen gifts funding from Tsinghua University, Institute for Interdisciplinary Information Sciences.

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

## Supplementary files

We provide a video illustrating our GUI tools, example task, and solutions. We also provide a tutorial for helping users to use our annotation system as a supplementary file.

## A How to choose keyframes

The selection of keyframes is determined by the annotators, as we believe humans can naturally decompose complex tasks into simpler ones for communication. When annotators manipulate a scene within the GUI, they can save the current scene as a key frame within the task representation. They can also add, modify, or delete key frames using the editors within the web server interface.

The annotators were instructed on what kind of decomposition would likely result in a working trajectory, i.e., segmenting the manipulation processes at contact point changes., which is a simple and effective way for the annotator to provide high-quality labels, without having any expert understanding of the physics and the solver. If the agent is required to manipulate a new object or the actuator needs to establish contact on a different face of the object, annotators would introduce a new keyframe. Additionally, we employ heuristic methods to segment YouTube videos. This segmentation occurs when there is a significant disparity between two frames, aiming to simplify the keyframe selection for annotators.

## B DSL for Optimization Programs

Table 2 lists the functions and primitives in our domain-specific language.

We rely on the large language model's few-shot ability to compile the natural language description like "lift up the white cube above the black wall" into an executable optimization program. The prompt start with the introduction to the DSL, listing the functions as well as the explanations, followed by several examples illustrating how to compose those functions to implement complex functions, e.g., by checking the $x$ coordinates of two objects to decide if one is on the left of the other. Then we provide several pairs of natural language inputs and corresponding programs. In the end, we provide the natural language input to compile. Below is an example prompt.

```
Here are the functions for obtaining objects and target objects. Their
    return types are all one single point-cloud object. A desc can
    only include strings that contain shape ["rope", "sphere", "box",
    "mat"], colors like ["white", "gray", "green", "red", "blue", "
    black"], and position like ["left", "right", "top", "bottom"]:
- get(desc) # Get the point cloud of the objects with the description,
     for example, get("left white mat"). If the desc is 'all', then
    get all the objects.
- goal(desc) # Get the point cloud of the goal for the objects with
    the given name. The name can also be 'all'.
- others(desc) # Get all other point clouds except the object with the
     given name.
- others_goals(desc) # Get all other point clouds of the goals except
    the object with the given name.

Here are the functions that use constraints. Note that the obj, and
    goal can only be one single point-cloud:
- keep(cond, start=s, end=t) # Minimize the reward from time s to time
     t. It only takes one cond as an argument.
- require(cond, end=t) # Reach the condition at time t. By default end
    =1.
- similar(obj, goal) # Compare the shape earth mover distance of the
    objects to the given goal shapes.
- touch(obj) # Minimize the distance between the tool and the object.
- fix_place(obj) # Ensure that the shape of the given object(s) and
    its positions do not change. It only takes one argument.
- no_break(obj) # Do not break the objects.
- away() # check if the actuator is away from all objects
```

| Category | FuncName | Explanation |
|---|---|---|
| Objects | get(desc) | Get the point cloud of the objects with the description, for example, get("left white mat"). If the desc is 'all', then get all the objects. |
| | goal(descr) | Get the point cloud of the goal for the objects with the given name. The name can also be 'all'. |
| Temporal conditions | keep(cond, start=0, end=1) | Minimize the cond from time s to time t. It only takes one cond as an argument. |
| | require(cond, end=1) | Reach the condition at time t. By default end is 1. |
| | and(cond1, cond2, ...) | Needs to satisfy all constraints |
| Shape operators | com(shape) | Compute the center of the mass of the point cloud |
| | similar(A, B) | Compare the shape distance using EMD distance |
| | pcd(shape) | Get the PyTorch tensor that represents the point cloud of the shape |
| | leftpart, rightend, etc. | Get a part of the point cloud based on the description. |
| Actuator | touch(ShapeA) | Minimize the signed distance function of the actuators and the shape |
| | away | Check if the actuator is far away from the shape |
| | roll, vertical, etc. | Get the rotation angle of the actuator |
| Tensor operators | Relation $>, <, \geq, \leq$ | Compare relationships of two vectors or scalar |
| | Algebra $+, -$ | Add values to something. |
| | $l_2$(A, B) | Compare the $l_2$ distance |
| | min, max | Get the min max of a tensor |
| Soft Body Constraints | fix_shape | Do not change the shape of the objects for too much |
| | fix_place | Keep the object not moving |
| | no_break | Do not break the objects |
| Special | stage(sample_fn, optimize_fn) | An optimization stage that may involve a motion planning procedure and an optimization procedure. It is only used for compiling. |
| | sample(tool_name, *conds) | Select the tool of tool_name. We provide "knife", "board", "rolling pin" and "gripper". The conds are list of conditions that we hope the tool satisfy |
| | optimize(*conds) | Optimize the trajectory to satisfy the conditions. |

Table 2: Elements in the optimization program

```
- roll() # actuators' rotation about x
- pitch() # actuator's rotation about y
- yaw() # actuator's rotation about y
- vertical() # actuator is vertical
- horizontal() # actuator is horizontal

Here are the example functions to obtain additional information:
- com(obj) # Center of the objects.
- pcd(obj) # point clouds of the objects.
- max(pcd(obj)) # max of the point cloud of the objects.
- min(pcd(obj)) # min of the point cloud of the objects.
- x(com(obj0)) # X coordinate of the points.
- x(min(pcd(obj0))) # smallest X coordinate of the boundary of the
    points.

We can compose those functions to implement various functions. Below
    are examples:
# The x coordinate of the objects
 x(com(obj0))

# Obj0 is on the left of obj1.
 lt(x(com(obj0)), x(com(obj1)))

# the front end of the Obj0 is above obj1.
 gt(y(com(frontend(obj0))), y(max(pcd(obj1))))

# Obj0 is in front of obj1.
 lt(z(com(obj0)), z(min(pcd(obj1))))

# deform the rope named by 'blue' into its goal shape
 require(similar('blue', goal('blue')))

# deform object 'blue' into its goal shape while fixing others
 require(similar('blue', goal('blue'))), keep(fix_place(others('blue')
    ))

# first do A then do B
 require(A, end=0.5), require(B, end=1.0)

# not rotate the actuator aboux x axis
 keep(roll() < 0.1)

Here are special function for
- stage(sample_clause, optimize_clause) # Each program may have
    multiple stage and each stage must have a 'sample' function and an
     'optimize' function. There are at most three stages. Start a new
    stage if and only if we need to change the actuators or manipulate
     different objects or parts.
- sample(actuator_name, *args) # sample the actuator with the given
    name in the beginning, args denotes for the conditions or
    requirements of the actuator's pose.
- optimize(*args) # conditions needs to satiesfied during the
    manipulation.

Below are examples of the scenes and the program to solve the scenes.

Input: grasp the front end of the blue rope vertically and then deform
     into its goal pose and please do not break it. Then grasp the
```

```
    back of the mat and then pick up the mat and move it into its goal
     position.
Program:
blue_rope = get("blue rope")
stage(
    sample("gripper", grasp(frontend(blue rope)), vertical()),
    optimize(
        require(similar(blue_rope, goal('blue rope'))),
        keep(touch(blue_rope)),
        keep(no_break(blue_rope))
    )
)
mat = get("mat")
stage(
    sample("gripper", grasp(backpart(mat))),
    optimize(
        require(l2(com(get(mat)), com(goal("mat")))),
        keep(touch(backpart(mat)))
    )
)

Input: cut the mat into its goal shapes and then move the knife away.
Program:
mat = get("mat")
stage(
    sample("knife", touch(mat)),
    optimize(
        require(similar(mat, goal('mat'))),
        keep(touch(mat), end=0.5),
        require(away())
    )
)

Input: move the object 'A' to the left of 'B' then move it to the
    right.
Program:
A = get('A')
B = get('B')
stage(
    sample("gripper", grasp(A)),
    optimize(
        require(lt(x(com(A)), x(com(B))), end=0.5),
        require(gt(x(com(A)), x(com(B))), end=1),
        keep(touch(A))
    )
)

Input: Put the board above all objects and deform them into their goal
     shapes and keep them not broken
Program:
stage(
    sample("board", gt(y("board"), y(max(pcd('all'))))),
    optimize(
        require(similar('all', goal('all'))),
        keep(touch('all')),
        keep(no_break('all')),
    )
)

Input: Use the gripper to grasp the right part of the object on the
    right and deform it into its target shape while fixing others.
Program:
```

```
left = get('left')
stage(
    sample("gripper", rightpart(left)),
    optimize(
        require(similar(left, goal('left'))),
        keep(fix_place(others('left'))),
        keep(touch(rightpart(left))),
    )
)

Input: There are two objects of different colors "red", "green". Use
    the board to touch them one by one. Deform them into their target
    shape.
Program:
stage(
    sample("board", touch("red")),
    optimize(
        keep(touch('red')),
        require(similar("red", goal("red")))
    )
)
stage(
    sample("board", touch("green")),
    optimize(
        keep(touch('green')),
        require(similar("green", goal("green")))
    )
)

Please compile the input into a program. The generated programs
    satisfy several requirements: Do not use functions not mentioned
    before; the program should be concise; only use 'touch' at most
    once within a single 'optimize' and if we need to manipulate
    multiple objects, please use touch('all').

Please include an 'optimize' for each stage. Do not use more than one
    stage if it does not need to manipulate different objects or parts
    . Here is the input to compile:
```

## B.1 Comparing GPT3.5 and GPT4

We notice that GPT4 has a better performance compared with GPT3.5. Below we provide the results of GPT3.5 and GPT4 of example tasks shown in the main text. Generally, we have found that it is common for GPT3.5 to misinterpret the instructions, add/ignore constraints, and write incorrect commands with incorrect grammar. In contrast, GPT4 can generate pretty good results.

Although GPT4 already outperforms GPT3.5 in the translation task, we observed several failure cases in the translation process. For example, the output of GPT4 will contain objects that do not exist. In a task that lifts a sphere above a box, the output may contain an invalid sentence like require(similar('sphere', goal('sphere above box'))) where 'sphere above box' which doesn't correspond to a proper object. It may also generate a statement with a type error. For example, sometimes it may output require(similar('sphere', y(max(pcd('box'))))), which would trigger a compilation error as we can not compute the shape distance between an object and a coordinate. Fortunately, those compilation errors could be detected during the compilation and can be resolved by resampling. However, there are also rare cases where LLM removes or adds more constraints to the instructions. For example, it may ask the solver to fix the position of an object that needs to move. In these cases, our solver may fail to find a suitable solution. Restricting the output to fit a particular syntax, for instance, using context-free grammars would resolve such problems.

**Test Case 1**   Grasp the front of the rope and then grasp the back of the rope. Deform the rope into its target shape.

GPT3.5: added constraints not mentioned in the language

```
rope = get("rope")
stage(
    sample("gripper", grasp(frontend(rope))),
    optimize(
        keep(touch(rope)),
        require(similar(rope, goal('rope'))),
        keep(no_break(rope))
    )
)
stage(
    sample("gripper", grasp(backpart(rope))),
    optimize(
        keep(touch(rope)),
        require(similar(rope, goal('rope'))),
        keep(no_break(rope))
    )
)
```

```
rope = get("rope")
stage(
    sample("gripper", grasp(frontend(rope))),
    optimize(
        keep(touch(frontend(rope))),
        require(similar(rope, goal("rope")), end=0.5)
    )
)
stage(
    sample("gripper", grasp(backend(rope))),
    optimize(
        keep(touch(backend(rope))),
        require(similar(rope, goal("rope")), end=1.0)
    )
)
```

**Test case 2** Use the gripper to first lift up the white cube above the black wall and then move it to the target location.

GPT3.5: it creates two stages for single a stage task (not need to regrasp the object).

```
white_cube = get("white cube")
black_wall = get("black wall")
stage(
    sample("gripper", lift(white_cube)),
    optimize(
        require(gt(y(com(white_cube)), y(max(pcd(black_wall))))),
        keep(touch(white_cube)),
        keep(no_break(white_cube))
    )
)
stage(
    sample("gripper", gt(x(com(white_cube)), x(com(black_wall)))),
    optimize(
        require(l2(com(white_cube), com(goal("white cube")))),
        keep(touch(white_cube)),
        keep(no_break(white_cube))
    )
)
```

```
white_cube = get("white cube")
black_wall = get("black wall")

stage(
    sample("gripper", grasp(white_cube)),
    optimize(
        require(gt(y(com(white_cube)), y(max(pcd(black_wall)))), end
            =0.5),
        require(l2(com(white_cube), com(goal("white cube"))), end=1),
        keep(touch(white_cube))
    )
)
```

**Test case 3**    Put the knife above the white object. Cut the knife down to match the target shape of the white object. Ensure that the knife does not rotate.

GPT3.5: It uses z instead y to extract the objectśheight while the latter is true. It also misinterprets the meaning to deform the knife into the goal.

```
white_obj = get("white")
stage(
    sample("knife", gt(z(com("knife")), z(max(pcd(white_obj))))),
    optimize(
        require(l2(com("knife"), com(white_obj))),
        require(similar("knife", goal("white"))),
        keep(pitch() < 0.1, yaw() < 0.1)
    )
)
```

GPT4

```
stage(
    sample("knife", above(get("white"))),
    optimize(
        require(similar(get("white"), goal("white"))),
        keep(touch("white")),
        keep(roll() < 0.1),
        keep(pitch() < 0.1),
        keep(yaw() < 0.1)
    )
)
```

**Test case 4**    Pick up the front part of object 'top left mat' and place it into the goal without breaking it.

GPT3.5: It adds an incorrect command to let the gripper to move away from the object in the beginning.

```
mat = get("top left mat")
stage(
    sample("gripper", grasp(frontpart(mat)), away()),
    optimize(
        require(similar(mat, goal('top left mat'))),
        keep(touch(mat)),
        keep(no_break(mat))
    )
)
```

GPT4

```
stage(
    sample("gripper", grasp(frontpart(get("top left mat")))),
    optimize(
        require(l2(com(get("top left mat")), com(goal("top left mat"))
            )),
        keep(touch(frontpart(get("top left mat")))),
        keep(no_break(get("top left mat")))
    )
)
```

## C    Reinforcement learning baseline

We use the SAC and PPO implementation from stable-baselines3 with hyperparameters in Table 3. Figure 8 compares the optimization performance of RL algorithms and the Adam optimizer with respect to the IOU score of the ending state with different numbers of sampled episodes for the 20 single-stage tasks of five categories.

Table 3: Parameters for Reinforcement Learning

| Table 4: SAC | | Table 5: PPO Parameters | |
|---|---|---|---|
| gamma | 0.95 | update steps | 2048 |
| learning rate | $3 \times 10^{-4}$ | learning rate | $3 \times 10^{-4}$ |
| buffer size | $10^{6}$ | entropy coef | 0 |
| target update coef | 0.005 | value loss coef | 0.5 |
| batch size | 256 | batch size | 64 |

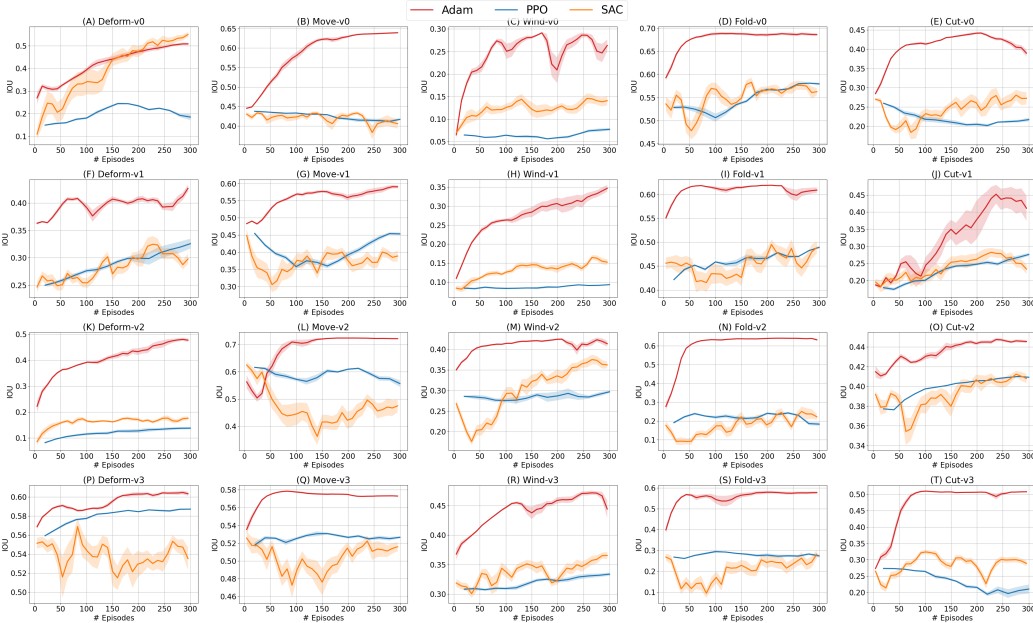

Figure 8: Comparison between RL and differentiable physics solver

## D    Evaluation metrics

We notice that in some tasks, a simple Intersection Over Union (IOU) score comparing shapes does not effectively explain the human intuition of task completion. For example, a solver might "cut" two

objects similar to the target shape without truly separating them, as the space between the two parts might only contain a minimal amount of space. In other circumstances, the solver may violate the requirements of no_break and break a rope into two or destroy a mat. To this end, we add checkers to check if two shapes are split and if a shape is not broken for success evaluation. For no-breaking constraint, we track particles and ensure their distance to their initial nearest neighbor does not change too much. On the other hand, we can check if two shapes are separated from each other by finding the connected components of particles.

