# OpenReview forum: "DiffVL: Scaling Up Soft Body Manipulation using Vision-Language Driven Differentiable Physics"
_NeurIPS.cc/2023/Conference — NeurIPS 2023 poster_

### Official Review · Reviewer_C43D · 2023-06-26

**Soundness:** 3 good
**Presentation:** 3 good
**Contribution:** 3 good
**Rating:** 6
**Confidence:** 3

**Summary:**

The paper works on trajectory generation for soft body manipulation with differentiable simulation. To address the key challenge of representing task goals for optimization, the authors propose to use natural language descriptions to lower the barrier for annotation, where a framework that utilizes LLM for translating natural language into optimization programs is developed to facilitate data collection.

**Strengths:**

The idea of task specification with natural language is an interesting idea;

The overall framework seems to be reasonable and technical sound, it is smart to leverage LLM for converting natural language descriptions into optimization programs;

The paper is well-presenting and easy to read.

**Weaknesses:**

More evaluations can be performed to validate the robustness of the trajectory generation;

There is no discussion of the failure cases;

Some missing citations in line 21;

Typo in figure 4: smapling -> sampling;

Some works that leverage natural language for task goal specifications, which may need to be discussed in the related work:

[1] StructFormer: Learning Spatial Structure for Language-Guided Semantic Rearrangement of Novel Objects, ICRA 2022;

[2] Differentiable Parsing and Visual Grounding of Natural Language Instructions for Object Placement, ICRA 2023

**Questions:**

In section 4.3, the authors say that a sampling-based RRT planner is used to determine the path for the actuator, and then the trajectory is refined with a gradient-based optimizer. Could the authors provide more details on how the trajectory is initially calculated and how it is refined for execution? Besides, it would be better to provide more snapshots for the rollouts to show the difference between the initial trajectory and the final ones;

Would the trajectory optimization be sensitive to the initialization? It would be interesting to conduct some experiments for demonstrating the robustness of the initialization strategy;

Are there any failure cases that are caused by the LLM translation? It would be good to understand the failure mode in which situations the LLM failed to generate appropriate optimization programs.

**Limitations:**

Limitations are discussed in the paper, but no failure cases are presented and analyzed.

---

> ### Author Rebuttal · Authors · 2023-08-09
>
> We thank you for your thorough assessment and positive feedback concerning our paper. It is genuinely rewarding to learn that you recognize the significance of our distinctive task representation, the novel method to integrate LLM, and the overall clarity evident in our writing. Your insights are invaluable to us.
>
>
> > Could the authors provide more details on how the trajectory is initially calculated and how it is refined for execution?
>
>
> We clarify that our approach does not use a gradient-based optimizer to optimize trajectories generated by the RRT planner. Motion planning and optimization occur in separate temporal phases. Initially, we plan and execute a trajectory to position the actuator at the specified initial pose. Subsequently, we initialize a new trajectory starting from this pose, and the actuator remains fixed in this trajectory as we initialize the policy with zero actions to maintain its position. We then only optimize the second trajectory to operate the actuator to complete the task. The optimizer does not touch the previous trajectory and keeps it unaltered.
>
>
> > More evaluations can be performed to validate the robustness of the trajectory generation & sensitivity to the initialization
>
> We emphasize that our vision-language task representation is designed to provide a good initial pose for trajectory optimization.  The `sample` function generates poses in line with the annotators' guidance, and the optimization trajectory starts with zero actions. This yields a robust optimizer for different optimization programs.
>
> The table below illustrates the solver's performance across several sampled tasks. The standard deviation, calculated across five seeds, highlights that the observed variance is not significant.
>
> |                   | Task 1 | Task 2 | Task 3 | Task 4 | Task 5 | Task 6 |
> | ----------------- | ------- | ------- | ------- | ------- | ------- | ------- |
> | Average IOU Score | 0.575   | 0.453   | 0.684   | 0.529   | 0.474   | 0.604   |
> | Variance          | 0.026   | 0.029   | 0.052   | 0.039   | 0.016   | 0.007   |
>
>
>
> > Discussion of the failure cases; failure cases caused by the LLM translation
>
> Although GPT4 already outperforms GPT3.5 in the translation task, we observed several failure cases in the translation process. For example, the output of GPT4 will contain objects that do not exist. In a task that lifts a sphere above a box, the output may contain an invalid sentence like `require(similar('sphere', goal('sphere_above_box')))` where `sphere_above_box` which doesn't correspond to a proper object. It may also generate a statement with a type error. For example, sometimes it may output `require(similar('sphere', y(max(pcd('box')))))`, which would trigger a compilation error as we can not compute the shape distance between an object and a coordinate. Fortunately, those compilation errors could be detected during the compilation and can be resolved by resampling. However, there are also rare cases where LLM removes or adds more constraints to the instructions. For example, it may ask the solver to fix the position of an object that needs to move. In these cases, our solver may fail to find a suitable solution. Restricting the output to fit a particular syntax, for instance, using context-free grammars would resolve such problems~[1].  We will conduct a more comprehensive analysis of these failure cases and elaborate on them in the revised manuscript.
>
> [1] Shin, Richard, et al. "Constrained language models yield few-shot semantic parsers." arXiv preprint arXiv:2104.08768 (2021).
>
> > Some works that leverage natural language for task goal specifications, which may need to be discussed in the related work
>
> Thanks for pointing out the related literature. Our work aligns closely with leveraging natural language for task goal specifications. Our vision-language task representation can be considered a richer extension of prior language goal representations in the following aspects:
> - Our vision-language task representation introduces a temporal dimension, offering a more detailed description of the *physical process* rather than just the *scene*. The description of the process aids the solver in synthesizing the trajectory beyond single-goal guidance.
> - We provide the GUI tool to directly create vision goals instead of only relying on language, yielding a more intuitive and accurate means of specifying goals.
> - This flexibility empowers us to focus on low-level physics and manipulate soft bodies effectively.
>
> We believe it is possible to integrate ideas from language-guided goal specifications into our work. We look forward to exploring the synergy between these approaches and will include more discussion in our revised manuscripts.
>
> We will improve our writing and polish the manuscripts. If you have any further questions, please do not hesitate to ask.

---

> > ### Author Response · Authors · 2023-08-17
> >
> > Dear reviewer,
> >
> > Thanks again for your suggestion to strengthen this work. As the rebuttal period is ending soon, we wonder if our response answers your questions and addresses your concerns. If yes, would you kindly consider raising the score? Thanks again for your very constructive and insightful feedback
> >
> > Best,
> >
> > Authors

---

> > ### Comment · Reviewer_C43D · 2023-08-18
> > **POST-REBUTTAL**
> >
> > Thanks to the authors for their updates. The supplementary technical details effectively address the inquiries I had, leading me to revise my rating to a "weak accept."

---

### Official Review · Reviewer_EcLH · 2023-07-07

**Soundness:** 3 good
**Presentation:** 3 good
**Contribution:** 3 good
**Rating:** 6
**Confidence:** 4

**Summary:**

This study focuses on soft-body manipulation problems such as flattening dough using a rolling pin, cutting deformable objects, and more. It introduces a method that engages non-expert users to provide detailed annotations and identify sub-goal states within key frames of the task video. Despite requiring additional human intervention, this approach generates a rich learning signal for manipulation tasks involving deformable objects. This significantly eases the learning difficulty associated with such tasks. The proposed method has been evaluated based on six fundamental deformable object manipulation skills, demonstrating its effectiveness.

**Strengths:**

This work proposes a unique representation of task specifications. It incorporates human-labeled annotations for deformable object tasks to shape an objective/reward function. Such a reward function provides far richer reward signals than just a goal state, potentially greatly reducing policy learning difficulties.

Furthermore, this study offers a new dataset that includes 100 deformable object manipulation tasks, complemented by human annotations on key frames. This contribution could be beneficial to the broader community and greatly aid studies related to deformable object manipulation.

The presentation of this research is both clear and organized.


**Weaknesses:**

Human annotation offers a valuable reward signal, which could potentially simplify the learning process. However, it also imposes certain constraints on the policy strategy. When states deviate from the demonstrated trajectory, adhering strictly to specific action commands could exacerbate the situation. Hence, at its current stage, it lacks the flexibility needed to handle unseen states not accounted for in the instructions/annotations.

Moreover, each task necessitates considerable human involvement in creating subgoal states and corresponding annotations. Although software solutions have been developed to assist in this process, it still represents a significant involvement of human resources and incurs considerable costs.


**Questions:**

Q1: How is a key frame chosen? Also, can you explain how the decomposition of long-horizon tasks occurs?


**Limitations:**

See weaknesses and questions.

---

> ### Author Rebuttal · Authors · 2023-08-09
>
> We sincerely appreciate your comprehensive evaluation and positive feedback on our paper. It's truly gratifying to know that you found value in our unique task representation, the novel manner in which we incorporated LLM, and the clarity throughout our writing.
>
> > Imposes constrains on policy strategies; Strict adherence to action commands can worsen trajectory deviations; Lacks flexibility for unseen states outside annotations.
>
>
> Thank you for bringing this issue to our attention. If we understand correctly, it refers to the case when our solver fails to reach a certain keyframe, making the reward signal in the subsequent stage useless. This might happen when users need to provide better annotations. We acknowledge it as a limitation of our current solver, and could in theory introduce a DAGGER-like approach to allow users to actively adjust annotations by adding or editing keyframes based on the found trajectory and offering corrective input. This integration of the annotation and optimization could provide a "closed-loop" control to help the agent recover from those unseen states.
>
>
> > Necessitates considerable human involvement
>
> Thank you for bringing this issue to our attention. While we recognize our current labeling system can be significantly improved, we would like to point out that our framework is already sufficient to help us build SoftVL100 --  a compact yet diverse dataset with many distinct tasks and high-quality annotations on how to achieve the desired manipulations. We believe considerable human involvement is a good thing and is synergetic with having an efficient annotation experience so that non-expert annotators can provide information-rich annotations for robotic tasks without unsatisfying overheads. We are confident that we can significantly reduce human efforts with enhanced toolchains. For example, using advanced tools such as Augmented Reality, 3D mice, and even language-based instructions can facilitate the operation of soft bodies. However, the further improvement of the toolchain lies in the realm of HCI and is beyond our scope. We would like to leave it as future work.
>
> > How is a key frame chosen? Also, can you explain how the decomposition of long-horizon tasks occurs?
>
> The selection of keyframes is determined by the annotators, as we believe humans can naturally decompose complex tasks into simpler ones for communication. When annotators manipulate a scene within the GUI, they can save the current scene as a key frame within the task representation. They can also add, modify, or delete key frames using the editors within the web server interface.
>
> The annotators were instructed on what kind of decomposition would likely result in a working trajectory, i.e., segmenting the manipulation processes at contact point changes., which is a simple and effective way for the annotator to provide high-quality labels, without having any expert understanding of the physics and the solver. If the agent is required to manipulate a new object or the actuator needs to establish contact on a different face of the object, annotators would introduce a new keyframe. Additionally, we employ heuristic methods to segment YouTube videos. This segmentation occurs when there is a significant disparity between two frames, aiming to simplify the keyframe selection for annotators.
>
>
> We hope our explanation help alleviate your concerns. If you have any further questions, please do not hesitate to ask.

---

> > ### Comment · Reviewer_EcLH · 2023-08-17
> > **Reply to Authors**
> >
> > Thanks for the rebuttal. My concerns have been addressed.

---

### Official Review · Reviewer_xC3q · 2023-07-07

**Soundness:** 4 excellent
**Presentation:** 4 excellent
**Contribution:** 4 excellent
**Rating:** 8
**Confidence:** 5

**Summary:**

This paper presents a novel approach to soft body manipulation employing the strengths of the Large Language Model (LLM). The key innovation is viewing tasks as data, with each data point consisting of an initial scene and an optimization objective. To tackle the challenge of task representation, the authors introduce DiffVL, a framework enabling non-expert users to define soft-body manipulation tasks to a differentiable solver using a combination of vision and language. The users specify tasks via an interactive simulator with a sequence of 3D scenes (keyframes) connected by natural language instructions. The authors also developed a corresponding GUI and curated SoftVL100, a vision-language dataset with 100 diverse tasks. They further developed a method that combines the power of a large-language model and differentiable physics to solve a wide variety of challenging long-horizon tasks in SoftVL100. This study's contributions lie in its new task representation, the developed GUI, the curated dataset, and the DiffVL method.

**Strengths:**

The paper under discussion is a robust, well-rounded work, bringing to the table significant contributions across multiple facets. It excels in terms of novel methodology, tool development and proposes a new dataset:

• The authors introduce an innovative multi-stage vision-language representation. This new approach simplifies the definition of soft-body manipulation tasks, making it accessible for non-expert user annotations - a noteworthy contribution to research in this area.

• In terms of tool development, the authors have crafted a corresponding Graphical User Interface (GUI), enhancing the user experience and overall accessibility of their proposed approach. They have also curated SoftVL100, a compilation of 100 realistic soft-body manipulation tasks derived from online videos. This dataset stands as a valuable resource for further research and application in this domain.

• Moreover, the authors have devised DiffVL, a method that bridges the gap between a large-language model and differentiable physics. This combination of strengths is adeptly applied to tackle a wide variety of challenging long-horizon tasks presented in SoftVL100, which marks a significant advancement in the field.

The experimental section of the paper is another key strength, featuring a thorough ablation study.


**Weaknesses:**

The relative weakness of the approach can be considered the amount of human labor required for dataset collection. Also, DSL could sometimes be too constrained to define arbitrary new tasks. Despite these points, they in no way diminish the overall quality of the work presented in the paper. The minor limitations identified simply highlight areas for potential future refinement. The project stands as an excellent example of research and tool development, and I am eager to see how it evolves.

**Questions:**

• It would be good to have more info about time constraints - simulation and training time for different tasks, not only training progress per epoch.
• IWere there any simulation failure cases? How challenging was simulator tuning for different tasks?


**Limitations:**

Limitations were addressed fairly well.

---

> ### Author Rebuttal · Authors · 2023-08-09
>
> We're thrilled about your positive feedback on our work. Your acknowledgment of our innovation, robustness, and solid foundation is truly gratifying. Your encouraging words inspire us to pursue excellence in all our efforts. Thank you.
>
> >  the amount of human labor required for dataset collection
>
> We agree that reducing the amount of human labor is an important research direction. We foresee that better UI design, although it is not our core contribution, can significantly improve the annotation process with more attention on the HCI frontier. For example, we can develop better tools like augmented reality (AR), 3D mice, and even language instruction. Our current workflow, involving keyframes and natural language, demonstrated how to construct the appropriate data format (namely, keyframes and natural language annotations), along with the UI system, which makes it feasible to both (a) enable end users to provide high-quality annotations and (b) having the data be readily usable by machines. We hope our work can bridge the realms of HCI and robot learning and bring better synergies between humans and robots.
>
> > DSL could sometimes be too constrained to define arbitrary new tasks.
>
> We acknowledge that our DSL is not designed to cover arbitrary natural language. For instance, our DSL assumes a fixed number of stages and often deals with one object per stage, making it less suitable for scenarios involving multiple objects moving simultaneously. Additionally, the requirement to specify exact time makes it challenging to implement temporal logic like `do A until B.` Moreover, the DSL supports only a limited range of spatial and dynamic relationships and lacks coverage for the various verbs used in everyday language.
>
> However, although rudimentary, our proposed DSL does not need to be "complete" as natural language. Notice that it serves as an auxiliary loss function for completing the correct trajectory, and the lack of certain DSL primitives does not make coming up with a trajectory infeasible but only more difficult. This difficulty can be overcome elegantly by further decomposing the task into sub-tasks, simply by providing more keyframes to guide the trajectory step by step, effectively making a motivated user always possible to solve the task with a physics backend.
>
> Moreover, it is possible to refine our DSL for a wider range of scenarios. An interesting approach is to conduct a more thorough user study as in [1], which can help us identify common elements and distill valuable insights to create a more refined DSL, ensuring that our DSL remains relevant and tailored to the needs of our users and the tasks they encounter.
>
> [1] Acquaviva, Sam, et al. "Communicating natural programs to humans and machines." Advances in Neural Information Processing Systems 35 (2022): 3731-3743.
>
> > Were there any simulation failure cases? How challenging was simulator tuning for different tasks?
>
> Our simulator is constructed using the MPM algorithm from PlasticineLab, which prioritizes time efficiency over simulation accuracy. Consequently, a frequent issue observed is that when the soft body moves at high speeds, it may penetrate through the rigid actuators due to the lower MPM resolution used for fast simulations. However, we did not yet fine-tune the simulator for specific tasks to keep the simulator tuning simple.
>
> >  It would be good to have more info about time constraints - simulation and training time for different tasks, not only training progress per epoch.
>
> For a single-stage task, it evenly takes 10 minutes for 300 training steps on a machine with NVIDIA GeForce RTX 2080 Super, which contains 80 steps in the simulation. For most tasks, 300 steps in training is sufficient. We will include a task-level task time analysis in our revised manuscript.

---

> > ### Comment · Reviewer_xC3q · 2023-08-18
> >
> > Thank you for the rebuttal. All my questions and concerns were addressed. I'll keep my score unchanged, and I think it's a solid and interesting work.

---

### Official Review · Reviewer_kids · 2023-07-07

**Soundness:** 3 good
**Presentation:** 3 good
**Contribution:** 2 fair
**Rating:** 4
**Confidence:** 4

**Summary:**

This paper demonstrates curating a set of 100 soft-body manipulation tasks and provides expert policies for them by using a mix of: annotators that provide supervision in the form of keyframes and/or natural langauge annotation, translating the annotations into programs via an LLM, and solving an optimization.

**Strengths:**

One strength of this paper is that obtaining manipulation policies for soft bodies is undoubtedly hard, and so anything that can do is has some potential value to the community.

Additionally, the dataset if released by the authors could be valuable.  The tasks and expert policies could be used as data to study methods that require some amount of expert data (imitation / offline RL / etc), and it's hard to both get these types of policies, as well as build simulation environments, in order to study these well. Good datatsets and tasks can be considered a bottleneck of manipulation research, especially with soft body manipulation, and the dataset could be valuable here.

**Weaknesses:**

The first sentence of this paper has unfinished citations. It reads, verbatim: "This paper focuses on soft body manipulation, a research topic with a wide set of applications such as folding cloth[cite], untangling cables [cite], and cooking foods[cite]."  This is a pity because it makes it hard to take the work too seriously given an incomplete first sentence.  Meanwhile, the rest of the paper seems relatively polished, so I'm doing my best to evaluate it seriously.

One limitation that should be kept in mind is that in multiple ways I only see the methods of this paper as being relevant to simulation, and not directly to the real world.  This applies both to the 3D GUI annotation process, as well as the process for converting into something that the differentiable physics solver can solve.  The work could help enable further studies in simulation that could eventually impact the real-world, indirectly either through helping provide evaluation tools or perhaps eventually sim-to-real, but not directly.

Overall, I think the main potential interesting thing is that the authors have actually provided expert policies for 100 or so soft-body manipulation tasks. This is hard to achieve.  I personally find that the existence of the policies and tasks is the main valuable and interesting thing, and less so the methods used to obtain them.

Additionally, 100 tasks is not that large for a crowdsourced effort.  This is an important point in my opinion because it's likely the process might have just gone faster if the first author designed all the tasks themselves rather than teaching others to do so.  One might need to get to 1,000 or 10,000 tasks in order to really start to see the benefits of a crowdsourced effort.  Accordingly this in my opinion undermines the care taken in the paper to make the annotation process usable by non-experts.

Another lens to look at this paper is: what actual results are shown?  There are two results tables/figures: Table 1, and Figure 7.  Table 1 shows that it is hard for SAC or PPO to solve the tasks, demonstrating that they are indeed not easy.  It also shows that their method, which has access to considerably more info than SAC/PPO due to its use of the annotations, can solve the tasks.  This is not a fully fair comparison, and that's fine.  The second results table shows that the multi-stage annotations and optimizations are useful.  This also makes sense.  It is hard to pinpoint what specific contribution claim this results figure supports though, since I don't think there is clear novelty in the multi-step framework presented by the authors.

**Questions:**

Do each of the points discussed in the weaknesses make sense?

**Limitations:**

The authors do discuss limitations in the conclusion. They should add that the methods are limited to simulation and impact on the real-world is left as a future question to address.
They don't discuss societal impacts but I think that's okay here.

---

> ### Author Rebuttal · Authors · 2023-08-09
>
> Thank you for your thoughtful evaluation. We genuinely appreciate your recognition of the significance of our soft body manipulation dataset and the challenges it presents. Your criticism is invaluable to us, and we would be more than happy to discuss your concerns in greater detail.
>
> > I personally find that the existence of the policies and tasks is the main valuable and interesting thing, and less so the methods used to obtain them.
>
> We want to emphasize that our contribution goes beyond just the dataset itself. Our work highlights that not only the dataset itself but also the task representation and methodology employed in creating the dataset play a crucial role in robot learning. With our approach, other researchers will be able to efficiently curate their own datasets in a similar manner, beyond what we have already collected.
> - We advocate the benefits of using crowd-sourcing for task collection, as opposed to conventional methods like gathering demonstration trajectories. The latter "requires non-trivial human efforts, is limited to robots with human-like morphologies, and may be challenging for collecting tasks that are non-trivial for humans." On the other hand, it also helps to avoid the tedious reward engineering and pave the path toward a large-scale dataset with diverse tasks.
> - We use a large language model to compile instructions for rewards, setting us apart from recent studies that use LLMs to generate high-level plans. This approach is not only more straightforward for the LLM but also allows us to leverage existing solvers for precise control over soft bodies, offering finer-grained manipulation capabilities.
> - Our approach introduces a novel way of representing tasks by combining vision and language, enabling the creation of a differentiable program capable of describing multi-stage temporal procedures. It simplifies trajectory annotation and seamlessly integrates with trajectory optimization; introducing the vision subgoal allows us to express complex deformations of soft bodies.
> - We also want to emphasize the importance and the potential of the GUI tool. It facilitates more straightforward communication between the robot and human annotators, enhancing the overall annotation process. The current GUI tool has some room for improvement -- for instance, using a 3D mouse to ease the manipulation of 3D objects.
>
> Through the tables/figures, we illustrated that "considerably more info" and "the multi-stage annotations and optimization" can solve the tasks, **which evidences the importance of suitable annotation and the design of the annotation tools**. This is why we designed the tool to help annotators gather and use that information to solve the tasks.
>
>
> > being relevant to simulation; they should add that the methods are limited to simulation and impact on the real-world is left as a future question to address.
>
> We acknowledge the gap between the simulator we used and real-world environments, yet we want to emphasize that we focus on both automatic tasks and policy generation. The goal is to develop a method to scale up the demonstration collection process and enable robots to master more diverse skills. Transferring the policies found by the differentiable simulator to the real world~[1] is important but orthogonal to our current study. Moreover, we believe that various components of our framework can offer valuable contributions to real-world scenarios: one such contribution lies in our vision language representation, which naturally represents tasks in the real world; leveraging large language models aids in generating rewards for evaluating real-world results; our GUI tool can effectively manipulate real-world materials, particularly in constrained tasks, serving as a direct interface between humans and robot controllers. Thus, we consider real-world experiments compatible with our approach instead of the limitation of the existing method.
>
> [1] Xu, Zhenjia, et al. "Roboninja: Learning an adaptive cutting policy for multi-material objects." arXiv preprint arXiv:2302.11553 (2023).
>
>
> > One might need to get to 1,000 or 10,000 tasks in order to really start to see the benefits of a crowdsourced effort. It's likely the process might have just gone faster if the first author designed all the tasks themselves
>
> We respectfully push back your point. Our framework has already benefited us by creating DiffVL100, the largest soft body manipulation task set so far. Notice how challenging it is for a single author to create 100 tasks, which may take $100\times 0.5=50$ hours, while involving several students, e.g., 10, in parallelizing the collection process can finish the time in one day ($50/10=5$ hours) and capable of generating a broader range of diverse tasks. Thus, we believe that crowdsourcing is vital in achieving this goal.  We have taken the first step to demonstrate how non-experts can contribute to scaling up robot learning, which may inspire the field to explore ways of involving individuals from various backgrounds in expanding robot manipulation capabilities. This direction is clearly promising and not constrained by the number of tasks. Scaling up to 1000 tasks requires in total $1000\times 0.5=500$ hours. It would be out of the capabilities of our group but can be readily adopted by other groups with better resources eventually.
>
>
> Once again, thank you for taking the time to review our manuscript. We will polish it further and include the discussions above. We're eager to discuss and address any concerns you may have.

---

> ### Author Response · Authors · 2023-08-18
>
> Dear reviewer,
>
> Thanks again for your suggestion to strengthen this work. As the rebuttal period is ending soon, we wonder if our response answers your questions and addresses your concerns. If yes, would you kindly consider raising the score? Thanks again for your very constructive and insightful feedback
>
> Best,
>
> Authors

---

> > ### Author Response · Authors · 2023-08-19
> >
> > Dear reviewer kids,
> >
> > Thanks again for your suggestion to strengthen this work. As the rebuttal period is ending soon, we wonder if our response answers your questions and addresses your concerns. If yes, would you kindly consider raising the score? Thanks again for your very constructive and insightful feedback!
> >
> > We are looking forward to your feedback!
> >
> > Best,
> >
> > Authors

---

> > > ### Author Response · Authors · 2023-08-20
> > > **Rebuttal ends tomorrow!**
> > >
> > > Dear reviewer,
> > >
> > > Thanks again for your suggestion to strengthen this work. As the rebuttal period is ending tomorrow, we wonder if our response answers your questions and addresses your concerns. If yes, would you kindly consider raising the score? Thanks again for your very constructive and insightful feedback
> > >
> > > Best,
> > >
> > > Authors

---

> > > > ### Author Response · Authors · 2023-08-21
> > > >
> > > > Dear reviewer kids,
> > > >
> > > > As the rebuttal period is ending today, we wonder if our response answers your questions and addresses your concerns. If yes, would you kindly consider raising the score? Thanks again for your very constructive and insightful feedback!
> > > >
> > > > Best,
> > > >
> > > > Authors

---

### Official Review · Reviewer_Scwt · 2023-07-08

**Soundness:** 4 excellent
**Presentation:** 3 good
**Contribution:** 3 good
**Rating:** 6
**Confidence:** 4

**Summary:**

This paper proposes DiffVL, a novel framework that tackles soft-body manipulation, which consists of a GUI for users to specify tasks easily and a large language model (LLM) for translating text instructions to programs for policy learning and execution.

**Strengths:**

- An intuitive user interface for task specification
- A novel combination of LLM and a differentiable simulator for program generation and optimization
- Good empirical success rate
- Clear writing

**Weaknesses:**

- Heavy user input needed on the magnitude of hours to specify task
- Comparison in Table 1 is a bit unfair since other pure RL baselines do not leverage language guidance
- Writing is not polished (e.g., missing citations in line 21)

**Questions:**

How does this approach differ from other recent robot learning papers utilizing LLM for instruction generation/high-level planning? The novelty in this aspect is obscure to me.

**Limitations:**

The authors discussed limitations adequately in the paper.

---

> ### Author Rebuttal · Authors · 2023-08-09
>
> We genuinely thank you for your thorough assessment and encouraging remarks regarding our manuscript. We are pleased to learn that you appreciated the intuitiveness of our user interface and the novelty of our approach to integrating LLM. Additionally, we are grateful for your recognition of our empirical study and the clarity of our writing.
>
> > Heavy user input needed on the magnitude of hours to specify task
>
> Thank you for bringing this issue to our attention. We acknowledge that minimizing human efforts is definitely an important research direction. However, most of the slowdowns can be significantly improved given more time and attention to details in the UI -- which is the realm of Human–computer interaction (HCI). For example, using advanced tools such as Augmented Reality, 3D mice, and even language-based instructions may accelerate the process. Our system can be much better with more attention to detail on the HCI front. However, we respectfully argue that while a better UI is relevant, it isn't the focus of our core contribution. Our current workflow, involving keyframes and natural language, serves as a guiding example for constructing UI systems that generate intuitive and machine-learnable data. This bridges the realms of HCI and robot learning and may inspire future research.
>
> > Comparison in Table 1 is a bit unfair since other pure RL baselines do not leverage language guidance
>
>
> Although we agree that it is worth integrating language into RL baselines, we would like to clarify that our RL baselines are set up in a very similar manner to the `-Optimize` approach. In both cases, we sample good actuator initialization poses and optimize policies with the sole objective of minimizing the shape distance. This setup ensures a fair comparison between our RL baselines and the `-Optimize` method. However, our RL baselines show significantly worse performance compared to the trajectory optimization baseline, which benefits from an analytical world model for optimization. This outcome is reasonable, given that previous work~[1] has already demonstrated the superiority of model-based trajectory optimization. As a result, we have focused our attention on conducting language-based experiments mainly on the optimization baseline, allowing us to concentrate on exploring the potential of our language-based approach.
>
> In addition, integrating language into RL baselines may entail non-trivial challenges. For example, many reward functions in our optimization program have temporal aspects, making the original state non-Markovian. As we focus on enhancing the optimization baseline through language-based experiments, we've chosen to defer tackling the complexities of language-conditioned RL to future research. We'll elaborate on this in the revised manuscript.
>
> [1] Huang, Zhiao, et al. "Plasticinelab: A soft-body manipulation benchmark with differentiable physics." arXiv preprint arXiv:2104.03311 (2021).
>
> > differ from other recent robot learning papers
>
> We acknowledge that several papers have emerged recently exploring the application of LLM in robots. However, at the time of submitting our paper, our method distinguished itself from other approaches in several key aspects. We would like to highlight the following points that make our work novel:
> - First, our method distinguishes itself from other language-conditioned policies by focusing on translating natural language into rewards rather than generating a direct plan, instructions, or actions, as in previous approaches such as R11, PALM-E, and CLIPort. The fundamental concept underlying our approach is that verifying if a generated trajectory accomplishes the task is inherently simpler than generating the right plan. Consequently, our method only requires the LLM to perform the task of compiling a reward program, which is considerably simpler compared to generating a plan, which may exceed the capabilities of the LLM.
> - Secondly, the ability to control low-level physics emerges after integrating a model-based trajectory optimizer, which differs from most existing work that mainly works on high-level actions like picking certain objects.
> - Thirdly, our research specifically targets soft body problems, whereas the majority of existing work primarily focuses on rigid bodies. Softbody manipulation is a unique domain where our approach of using both vision and language guidance as a way of communicating a task is best suited. In comparison, in a rigid body setting, a language-only task definition such as "put the cup on the table" might be already sufficient.
> - Lastly, it is crucial to emphasize that our framework has a distinctive objective of collecting tasks as data, which sets it apart from existing work that assumes the existence of pre-defined tasks. This novel formulation enables the injection of human feedback into the trajectory generation process, providing a more interactive and adaptive approach. By incorporating human feedback and iteratively refining the system, we aim to create a more robust and flexible framework that can adapt to a wide range of scenarios and tasks.
>
> > Writing is not polished (e.g., missing citations in line 21)
>
> Thanks for pointing out the issue. We will publish it carefully.
>
>
> We hope our explanation helps alleviate your concerns. If you have any further questions, please do not hesitate to ask.

---

> > ### Comment · Reviewer_Scwt · 2023-08-14
> > **Thank you for your response**
> >
> > I appreciate the detailed response. My concerns are addressed well.

---

### Author Rebuttal · Authors · 2023-08-09

# General response

We thank all reviewers and ACs for their time and effort in reviewing the paper. We are glad that the reviewers generally recognized the following contributions.

**Problem and dataset** The paper curated a hard and valuable dataset (`kids`, `xC3q`), which is beneficial to the broader community and greatly aids studies related to deformable object manipulation (`EcLH`).

**Method** The paper proposes a novel framework combining LLM and a differentiable simulator(`Scwt`). The multi-stage vision-language representation is innovative (`xC3q`), interesting (`C43D`) and unique (`EcLH`). The overall framework is reasonable and technically sound (`C43D`). It is smart to leverage LLM(`C43D`), and the reward signal is valuable(`EcLH`).

**Tool development** Intuitive interface (`Scwt`) and GUI that enhances the user experience (`xC3q`).

**Experiments** The paper achieved a good empirical success rate (`Scwt`) and provided a thorough ablation study (`xC3q`).

**Presentation** The writing is clear (`Scwt`). The paper is organized (`EcLH`) and easy to read (`C43D`).

We have carefully considered all the questions and concerns raised by the reviewers and provided a detailed response to each reviewer's questions and concerns. We have polished our manuscripts and fixed typos.

We hope our responses have convincingly addressed all reviewers’ concerns. We thank all reviewers’ time and efforts again! Please don’t hesitate to let us know of any additional comments on the manuscript or the changes.

Best Regards,

Authors

---

### Author Response · Authors · 2023-08-12
**Thank you and we are looking forward to your post-rebuttal feedback!**

Dear AC and all reviewers:

Thanks again for all the insightful comments and advice, which helped us improve the paper's quality and clarity.

The discussion phase has been on for several days and we have not heard any post-rebuttal responses yet.

We would love to convince you of the merits of the paper. Please do not hesitate to let us know if there are any additional experiments or clarification that we can offer to make the paper better. We appreciate your comments and advice.

Best,

Authors

---

### Decision · Program_Chairs · 2023-09-21

**Decision:**

Accept (poster)

**Comment:**

This submission proposes a novel method related to soft-body manipulation using LLM. The submission also provides a GUI to specify tasks and also a vision-langauge soft-body manipulation dataset too. While the submission proposes a rather heavy engineering system method, it questions a timely important problem. However, there are several legitimate concerns from reviewers: if it is scalable to the real world (both the method and the gui), and also if the provided dataset is really useful in terms of its scale. Summing them up, we believe it is good to be shared to the community, but if the authors wants this paper to be more impactful, we suggest to increase the dataset size at the least.